# Synthesis of Ethynyl Trifluoromethyl Sulfide and Its Application to the Synthesis of CF_3_S-Containing Triazoles

**DOI:** 10.3390/molecules30112358

**Published:** 2025-05-28

**Authors:** Alejandra Riesco-Domínguez, Hussein Hammoudeh, Daniel Blanco-Ania, Floris P. J. T. Rutjes

**Affiliations:** Institute for Molecules and Materials, Radboud University, Heyendaalseweg 135, 6525 AJ Nijmegen, The Netherlandsdaniel.blancoania@ru.nl (D.B.-A.)

**Keywords:** (trifluoromethyl)sulfanyl, (trifluoromethyl)thio, ethynyl trifluoromethyl sulfide, triazoles, click chemistry

## Abstract

The unprecedented use of ethynyl trifluoromethyl sulfide (CF_3_S–C≡CH) as a synthetically useful building block has been described for the first time. It was reacted with various aromatic and aliphatic azides under copper-catalyzed conditions to yield a novel class of 1,4-disubstituted triazoles bearing the SCF_3_ group (15 examples, up to 86% yield).

## 1. Introduction

The incorporation of the (trifluoromethyl)sulfanyl group (SCF_3_) into druglike molecules may significantly enhance their bioavailability and induce more effective transport through lipid membranes (Hansch parameter π = 1.44) [1,2]. Consequently, introducing the SCF_3_ group into organic molecules is relevant for agrochemical and pharmaceutical communities that focus on isostere-based drug design [3]. In this regard, synthetic chemists also aim at developing new pathways for the incorporation of this SCF_3_ substituent, hence contributing to discoveries in biomedical research [4]. A few analogs derived from biologically active compounds have been developed since the chemistry of the SCF_3_ group became more widespread (Figure 1). For example, the SCF_3_ group has been used as an isostere in losartan analogs, used clinically for the treatment of cardiovascular diseases [5]. Vaniliprole and JKU 0422 were developed as analogs of the insecticide fipronil, a compound with a trifluoromethyl sulfoxide [6]. Finally, a methionine analog was also developed, which is of interest for peptide design because of the strong steric and electrostatic properties of the CF_3_ group [7].

The use of chemical building blocks represents one of the major strategies of the pharmaceutical and agrochemical industries for the construction of compound libraries. Thus, the discovery of new CF_3_S-containing building blocks that can be utilized for the synthesis of potentially biologically active molecules represents an important focus for organic chemists nowadays. In this regard, the majority of known building blocks bearing the SCF_3_ group consist of either (hetero)aromatic or disubstituted vinylic/acetylenic molecules [8,9,10,11]. Surprisingly, however, the synthesis of the parent building blocks CF_3_S-ethene/acetylene (**2** and **3**) and their application to prepare heterocycles are unexplored. We have reported the synthesis of the trifluoromethyl vinyl sulfide building block **2** from chloro alkane **1** and its application to the synthesis of a new class of isoxazolidines **4** by 1,3-dipolar cycloaddition reactions with nitrones (Figure 1) [12]. The inherent potential of building block **2** [13] motivated us to study the synthesis of its CF_3_S-acetylene counterpart **3** as well. Thus, we report herein the first synthesis of ethynyl trifluoromethyl sulfide **3** [14] and its use as a building block for the synthesis of drug-like CF_3_S-containing 1,4-disubstituted triazoles **5** (Figure 1).

## 2. Results and Discussion

The synthesis of CF_3_S–C≡CH (**3**) started with the preparation of trifluoromethyl vinyl sulfide **2** from commercially available chloro alkane **1** (Figure 2).

We applied the conditions reported by us (1.3 equiv of KO*^t^*Bu, 1.0 M solution in THF) [12] to afford alkene **2** quantitatively. Then, bromine was added to the solution containing alkene **2** [15]. Initial experiments showed that the addition of 2.0 equiv of Br_2_ gave full conversion to dibromide **6**, but that 4-bromobutan-1-ol was also formed as a side product from the reaction of bromine with THF [16,17]. We therefore replaced THF with CH_2_Cl_2_ (except for the stock solution of KO*^t^*Bu in THF) and reduced the amount of Br_2_ to avoid side product formation. Alkene **2** was then prepared from chloro alkane **1** in CH_2_Cl_2_ (KO*^t^*Bu, 1.3 equiv), with full conversion under these conditions. The subsequent addition of Br_2_ (1.1 equiv) afforded dibromide **6** with full conversion and without side products (Figure 2). We tried to isolate dibromide **6** by distillation, but these attempts resulted in the decomposition of **6** when evaporating the solvent.

Finally, the last steps of the synthesis of CF_3_S–C≡CH (**3**) implied a double elimination reaction of dibromide **6** (Table 1). It is worth noting that compounds **2**, **6**, **7** and **3** were not isolated and that the reactions were followed in all cases by ^1^H and ^19^F NMR. Preliminary studies using 1,2-dibromoethyl phenyl sulfide as a model substrate showed that KO*^t^*Bu, NaOH and NaNH_2_ (in excess) were not suitable bases for both eliminations to occur. After screening several other bases, we concluded that the combination of KO*^t^*Bu and NaHMDS gave the best results (Table 1).

The reaction of dibromide **6** with NaHMDS (2.0 equiv, entry 1) afforded bromo alkene **7** with full conversion after 17 h of reaction time. The addition of one more equivalent revealed the disappearance of alkene **7** by ^1^H NMR. Assuming that alkyne **3** was formed in the reaction, and based on previous results from our research group that revealed the importance of high pressure to promote the cycloaddition reaction of alkene **2** with nitrones [12], 1-azido-4-nitrobenzene was added and the reaction mixture was brought under 15 kbar of pressure [18,19,20]. Unfortunately, no product was obtained and the starting azide was recovered, so we continued investigating the formation of alkyne **3**. We repeated the same procedure, confirming that alkene **7** was formed after 1.5 h and with the addition of 2.0 equiv of NaHMDS (entry 2). Unfortunately, the addition of one further equivalent of NaHMDS did not result in the formation of alkyne **3**. When we used KO*^t^*Bu as the base for the first elimination (1.5 equiv, entry 3), we observed the formation of not only alkene **7** but also alkyne **3** (ratio **7**/**3** 4:1). The subsequent addition of NaHMDS (0.6 equiv) increased the formation of alkyne **3**, even though alkene **7** was still present in the reaction mixture. When we used KO*^t^*Bu and NaHMDS (1.3 and 0.7 equiv, respectively; entry 4), an increase in alkyne **3** was obtained. Finally, the best result was obtained by employing KO*^t^*Bu (1.3 equiv) and a subsequent excess of NaHMDS (1.7 equiv), providing alkyne **3** as the sole product in the reaction mixture (entry 5).

With CF_3_S–C≡CH (**3**) in hand, we focused our attention on the azide–alkyne cycloaddition reaction [21,22,23,24,25,26,27] for the synthesis of 1,4-disubstituted triazoles (**5**) bearing the SCF_3_ group [28]. Triazoles are some of the most exploited structures in heterocyclic chemistry because their structural motif occurs in products with a diversity of biological activity, such as antiviral, analgesic, anti-inflammatory, anticonvulsant, antimicrobial and anticancer effects [29,30,31,32].

Following the same strategy described by us for the synthesis of CF_3_S-containing isoxazolidines [12], we studied the high-pressure-promoted synthesis of CF_3_S-containing 1,4-disubstituted triazoles **5** (Table 2).

We first performed the reaction of azide **8a** (R = NO_2_) and CF_3_S–C≡CH (**3**) at 21 °C under 15 kbar of pressure (entry 1). Promisingly, the reaction proceeded smoothly and, after 72 h reaction time, we isolated triazole **5a** in a 78% yield. Surprisingly, compound **5a** was the only regioisomer formed [33], representing the first non-catalyzed regioselective azide–alkyne 1,3-dipolar cycloaddition reported in the literature. Azide **8a**, however, did not show any conversion into the desired triazole **5a** under thermal conditions (30 °C; entry 2) for 72 h. Initially, we refrained from using higher temperatures because of the volatility of CF_3_S–C≡CH (**3**). The reaction times for azides **8b** (R = F, entry 3) and **8c** (R = CF_3_, entry 4) were longer (120 h led to 2:7 and 4:7 ratios for **5b**/**8b** and **5c**/**8c**, respectively) and the yields were rather low (49% and 38%, respectively), even when heated up to 50 °C. Applying the same reaction conditions to substrates **8d** (R = Me, entry 5) and **8e** (R = OMe, entry 6), the ratios of **5d**/**8d** and **5e**/**8e** were 1:6 and 1:15, respectively, after 120 h of reaction time. As a result, we decided to perform the synthesis of triazoles **5** under copper-catalyzed conditions. Thus, the cycloaddition of azide **8a** with CF_3_S–C≡CH (**3**), CuSO_4_·5H_2_O (0.01 equiv), sodium ascorbate (0.02 equiv) and benzoic acid (0.1 equiv), in a mixture of *^t^*BuOH/H_2_O/CH_2_Cl_2_ at 30 °C, afforded triazole **5a** after 16 h in an 82% yield (entry 7). Compound **5b** was isolated in an 85% yield when using azide **8b** (R = F) under copper-catalyzed conditions at 60 °C (entry 8). These conditions greatly improved the results as compared to the high-pressure-promoted reactions and reduced the reaction times considerably.

Various azides were examined in the copper-catalyzed azide–alkyne cycloaddition reaction [34] by employing the conditions shown in entries 7 and 8 of Table 2. We used commercially available azides (**8b**–**g**, **8n** and **8o**) and freshly synthesized azides **8a** and **8h**–**m** (using the corresponding aniline under the classical conditions [35]: NaNO_2_ and TsOH followed by reaction with NaN_3_; see Section 3.2) as substrates. A total of 12 aromatic azides (**8a**–**l**) with electron-donating or electron-withdrawing groups at the 2-, 3- and 4-positions were used to study the scope of the copper-catalyzed cycloaddition reaction with CF_3_S–C≡CH (**3**; Figure 3). These reactions proceeded at different temperatures and reaction times and in varying yields (all isolated yields) depending on the position and electronic effect of the aryl substituents of azides **8a**–**l**. The yields were moderate to very good (61–86%; **5a**–**d**, **5f**, **5g** and **5l**), except when the phenyl ring was substituted with a MeO or CN group (**5e**, **5h**–**5k**). Most of the azides reacted at 50 °C, yielding triazoles **5c**, **5e** and **5h**–**l**. Compounds **5a**, **5f** and **5g** were formed at 30 °C, whereas triazoles **5b** and **5d** required 60 and 70 °C, respectively.

The reaction times also varied with the substrate. The majority of the reactions were carried out for 48 h (compounds **5d**, **5f** and **5h**–**l**), whereas the reaction rates were accelerated (16 h) for compounds **5a**–**c** and **5g**. For compounds **5e**, **5h** and **5j**, bearing a methoxy substituent at the 4-, 3- and 2-positions of the phenyl ring, we identified the formation of the corresponding 1,5-disubstituted regioisomers by NMR analysis (^1^H, ^13^C and HSQC) in 1,4/1,5 ratios of 10:1 (for compounds **5h** and **5j**) and 12:1 (for compound **5e**) [36,37,38].

Finally, we expanded the scope of this reaction to the synthesis of heteroaromatic and aliphatic CF_3_S-containing triazoles (Figure 3). In this manner, we were able to synthesize pyridinyl triazole **5m** after 16 h of reaction time at 70 °C in a 65% yield. Moreover, aliphatic triazoles (**5n** and **5o**) were synthesized in good yields (62 and 70%) at 50 °C for 16 h (Figure 3).

## 3. Materials and Methods

### 3.1. Reagents and Equipment

Reagents were obtained from commercial suppliers and were used without purification. Standard syringe techniques were applied for the transfer of dry solvents and air- or moisture-sensitive reagents. Reactions were followed, and *R*_F_ values were obtained, using thin layer chromatography (TLC) on silica gel-coated plates (Merck 60 F254) with the indicated solvent mixture. Detection was performed with UV light and/or by charring at ca. 150 °C after dipping into a solution of either 2% anisaldehyde in ethanol/H_2_SO_4_, KMnO_4_ or ninhydrin. Infrared spectra were recorded on an IR-ATR Bruker TENSOR 27 spectrometer. High-resolution or accurate mass measurements (ΔM < 3 mmu or 5 ppm) were recorded on a JEOL AccuTOF-CS JMS-T100CS for electrospray (spectra recorded in infusion in MeOH containing 50nM PPG-475 as internal mass-drift compensation standard) or a JEOL AccuTOF-GCv JMS-T100GCv (GC/electron ionization MS, column bleeding at high temperature used as internal mass drift compensation standard). NMR spectra (see Appendix A) were recorded at 298 K on a Varian Inova 400 (400 MHz), Bruker Avance III 400 MHz or Bruker Avance III 500 MHz spectrometer in the solvent indicated. Chemical shifts are given in parts per million (ppm) with respect to tetramethylsilane (0.00 ppm) as an internal standard for ^1^H NMR and to CDCl_3_ (77.16 ppm) as an internal standard for ^13^C NMR. Coupling constants are reported as *J* values in hertz (Hz). ^1^H NMR data are reported as follows: chemical shift (ppm), multiplicity (s = singlet, d = doublet, dd = doublet of doublets, dt = doublet of triplets, ddd = doublet of doublet of doublets, dtdq = doublet of triplet of doublet of quartets, dq = doublet of quartets, ddd = doublet of doublet of doublets, ddt = doublet of doublets of triplets, dddd = doublet of doublets of doublets of doublets, ddquint = doublet of doublet of quintets, dddquint = doublet of doublet of doublet of quintets, quint = quintet, t = triplet, td = triplet of doublets, tt = triplet of triplets, m = multiplet, b = broad), coupling constants (Hz), integration and assignment. Compounds were fully characterized by ^1^H and ^13^C spectra and 2D COSY, HSQC, HMBC, NOESY and HOESY spectra. Column or flash chromatography was carried out using ACROS silica gel (0.035–0.070 mm, 60 Å pore diameter).

### 3.2. General Procedure for the Synthesis of Aromatic Azides 8h–k

The corresponding aniline **9h**–**k** (1.0 mmol; Figure 4) was added to a solution of TsOH·H_2_O (1.62 g, 9.0 mmol) in H_2_O (9 mL). After stirring for 1 min, anhydrous NaNO_2_ (0.621 g, 9.0 mmol) was added gradually over 5 min. The resulting solution was then stirred for a period between 2 and 60 min until the starting amine disappeared (reactions were monitored by TLC). Anhydrous NaN_3_ (0.104 g, 1.6 mmol) was added to the resulting solution, and the immediate emission of N_2_ was observed. The solid aromatic azides (**8h** and **8i**) were filtered off, washed with H_2_O (50 mL) and dried in vacuo, whereas the oily azides (**8j** and **8k**) were extracted with AcOEt (3 × 10 mL) and dried over Na_2_SO_4_. The suspension was filtered off and dried under reduced pressure.

#### 3.2.1. 1-Azido-3-methoxybenzene (**8h**) [39]

According to the general procedure, the reaction of aniline **9h** (123 mg, 1.0 mmol) afforded azide **8h** (144.7 mg, 0.97 mmol). **^1^H NMR** [400 MHz, δ (ppm), CDCl_3_]: 7.25 (t, *J* = 8.1 Hz, 1 H), 6.67 (ddd, *J* = 8.3, 2.4, 0.8 Hz, 1 H), 6.65 (ddd, *J* = 8.0, 2.1, 0.9 Hz, 1 H), 6.55 (t, *J* = 2.2 Hz, 1 H), 3.80 (s, 3 H). **Yield**: 97%.

#### 3.2.2. 3-Azidobenzonitrile (**8i**) [40]

According to the general procedure, the reaction of aniline **9i** (118.1 mg, 1.0 mmol) afforded azide **8i** (136.9 mg, 0.95 mmol). **^1^H NMR** [400 MHz, δ (ppm), CDCl_3_]: 7.65–7.56 (m, 2 H), 7.28–7.25 (m, 1 H), 7.22 (td, *J* = 7.7, 1.0 Hz, 1 H). **Yield**: 95%.

#### 3.2.3. 1-Azido-2-methoxybenzene (**8j**) [39]

According to the general procedure, the reaction of aniline **9j** (123.2 mg, 1.0 mmol) afforded azide **8j** (141.7 mg, 0.95 mmol). **^1^H NMR** [400 MHz, δ (ppm), CDCl_3_]: 7.10 (td, *J* = 7.8, 1.7 Hz, 1 H), 7.02 (dd, *J* = 7.8, 1.7 Hz, 1 H), 6.97–6.88 (m, 2 H), 3.88 (s, 3 H, OC*H*_3_). **Yield**: 95%.

#### 3.2.4. 2-Azidobenzonitrile (**8k**) [41]

According to the general procedure, the reaction of aniline **9k** (118.1 mg, 1.0 mmol) afforded azide **8k** (115.3 mg, 0.80 mmol). **^1^H NMR** [400 MHz, δ (ppm), CDCl_3_]: 7.50–7.40 (m, 2 H), 7.30–7.24 (m, 2 H). **Yield**: 80%.

Azides **8b**–**g** were commercially available, and azides **8a** [35], **8l** [42] and **8m** [43] were previously prepared in our research group according to procedures published in the literature.

### 3.3. General Procedure for the Synthesis of Ethynyl Trifluoromethyl Sulfide (**3**)

In a sealed vial under a nitrogen atmosphere, 2-chloroethyl trifluoromethyl sulfide (**1**; 100 mg, 0.608 mmol) was dissolved in CH_2_Cl_2_ (1 mL). The solution was cooled to 0 °C and subsequently KO*^t^*Bu (790 µL, 0.790 mmol, 1.3 equiv, 1.0 M solution in THF) was slowly added. The reaction mixture was stirred at 21 °C for 90 min to form trifluoromethyl vinyl sulfide (**2**). Then, Br_2_ (34 µL, 107 mg, 0.668 mmol, 1.1 equiv) in CH_2_Cl_2_ (1 mL) was added at 21 °C to the solution containing alkene **2**. The reaction mixture was stirred for 1 h, until the orange color of the mixture changed to a pale-yellow color, to afford the CF_3_S-dibromo derivative **6**. Subsequently, the mixture was filtered and the CH_2_Cl_2_ was evaporated under reduced pressure to reduce the volume of the mixture by 50%. The reaction mixture was then cooled to 0 °C, KO*^t^*Bu (790 µL, 0.790 mmol, 1.3 equiv, 1.0 M solution in THF) was slowly added, and the reaction mixture was stirred at 21 °C for 90 min to afford alkene **7**. Finally, NaHMDS (517 µL, 1.034 mmol, 1.7 equiv, 2.0 M solution in THF) was slowly added and the mixture stirred for 1 h to give CF_3_S–C≡CH (**3**). ^1^H NMR and ^19^F NMR were checked after every reaction step and used for the final characterization of CF_3_S–C≡CH (**3**).

#### 3.3.1. Trifluoromethyl Vinyl Sulfide (**2**) [12]

**^1^H NMR** [400 MHz, δ (ppm), THF-*d*_8_]: 6.54 (dd, *J* = 16.5, 9.4 Hz, 1 H, 1-C*H*), 5.72 (dq, *J* = 9.4, 1.5 Hz, 1 H, 2-C*H*_a_), 5.70 (d, *J* = 16.5, 1 H, 2-C*H*_b_). **^13^C NMR** [101 MHz, δ (ppm), THF-*d*_8_]: 129.8 (q, *J* = 306.5 Hz, S*C*F_3_), 124.4 (q, *J* = 1.0 Hz, 2-*C*), 121.3 (q, *J* = 3.2 Hz, 1-*C*). **^19^F NMR** [377 MHz, δ (ppm), THF-*d*_8_]: −43.6.

#### 3.3.2. 1,2-Dibromoethyl Trifluoromethyl Sulfide (**6**)

**^1^H NMR** [400 MHz, δ (ppm), CDCl_3_]: 5.46 (dd, *J* = 7.1, 5.7 Hz, 1 H, 1-C*H*), 4.00 (ddq, *J* = 11.4, 5.8, 0.5 Hz, 1 H, 2-C*H*H), 3.91 (ddq, *J* = 11.4, 7.1, 0.6 Hz, 1 H, 2-CH*H*). **^13^C NMR** [126 MHz, δ (ppm), CD_2_Cl_2_/THF-*d*_8_]: 130.37 (q, *J* = 309.1 Hz, S*C*F_3_), 48.3 (1-*C*), 37.0 (2-*C*). **^19^F NMR** [377 MHz, δ (ppm), CDCl_3_]: −40.9.

#### 3.3.3. 1-Bromovinyl Trifluoromethyl Sulfide (**7**)

**^1^H NMR** [500 MHz, δ (ppm), CD_2_Cl_2_/THF-*d*_8_]: 6.56 (d, *J* = 2.3 Hz, 1 H, 2-C*H*H), 6.41 (dq, *J* = 2.3, 0.8 Hz, 1 H, 2-CH*H*). **^13^C NMR** [126 MHz, δ (ppm), CD_2_Cl_2_/THF-*d*_8_]: 138.2 (q, *J* = 1.3 Hz, 2-*C*), 130.0 (d, *J* = 310.5 Hz, S*C*F_3_), 112.3 (q, *J* = 2.9 Hz, 1-*C*). **^19^F NMR** [377 MHz, δ (ppm), CDCl_3_]: −42.5.

#### 3.3.4. Ethynyl Trifluoromethyl Sulfide (**3**)

**^1^H NMR** [400 MHz, δ (ppm), CDCl_3_]: 3.33 (s, 1 H, C*H*). **^19^F NMR** [377 MHz, δ (ppm), CDCl_3_]: −43.2.

### 3.4. General Procedure for the Synthesis of 1,4-Disubstituted-1H-1,2,3-Triazoles **5a–o**

A solution containing a mixture of CF_3_S–C≡CH (**3**; 3.0 equiv) and the corresponding azide **8** (1.0 equiv) in CH_2_Cl_2_/THF was added to a solution of CuSO_4_·5H_2_O (2.5 mg, 0.01 mmol), sodium ascorbate (4 mg, 0.02 mmol) and benzoic acid (12 mg, 0.1 mmol) in *^t^*BuOH/H_2_O (1:2 *v*/*v*, 1.0 mL) in a 4 mL vial. Then, CH_2_Cl_2_ was added in order to fill the vial completely. The resultant mixture was stirred for the stated time at the indicated temperature for every reaction (reactions were followed by ^1^H and ^19^F NMR). The reaction mixture was then quenched with H_2_O (20 mL) and extracted with CH_2_Cl_2_ (3 × 15 mL). The combined organic layers were washed with H_2_O and brine, dried over anhydrous Na_2_SO_4_, filtered off and concentrated in vacuo. The crude product was purified by column chromatography (heptane/AcOEt, 4:1) to afford the corresponding triazoles **5a**–**p**.

#### 3.4.1. 1-(4-Nitrophenyl)-4-[(trifluoromethyl)sulfanyl]-1*H*-1,2,3-triazole (**5a**)

According to the general procedure, the reaction of 1-azido-4-nitrobenzene **8a** (21 mg, 0.13 mmol) with CF_3_S–C≡CH (**3**) at 30 °C for 16 h afforded triazole **5a** (46.4 mg, 0.16 mmol) as a yellow–brown solid. **^1^H NMR** [400 MHz, δ (ppm), CDCl_3_]: δ 8.51–8.44 (m, 2 H, 3-C*H* + 5-C*H*), 8.42 (s, 1 H, C*H*), 8.05–7.98 (m, 2 H, 2-C*H* + 6-C*H*). **^13^C NMR** [101 MHz, δ (ppm), CDCl_3_]: 148.0 (4-*C*), 140.5 (1-*C*), 132.0 (*C*–S, indirect observation), 128.33 (*C*H), 128.32 (q, *J* = 310.0 Hz, S*C*F_3_), 125.9 (3-*C* + 5-*C*), 121.1 (2-*C* + 6-*C*).**^19^F NMR** [377 MHz, δ (ppm), CDCl_3_]: −42.5. **FTIR** [$\bar{\nu }$ (cm^−1^)]: 2924, 1599, 1519, 1348, 1144, 1102, 907, 733. ***R*_F_**: 0.32 (heptane/AcOEt, 4:1). **Yield**: 82%.

#### 3.4.2. 1-(4-Fluorophenyl)-4-[(trifluoromethyl)sulfanyl]-1*H*-1,2,3-triazole (**5b**)

According to the general procedure, the reaction of 1-azido-4-fluorobenzene **8b** (250 µL, 0.125 mmol, 0.5 M solution in *^t^*BuOMe) with CF_3_S–C≡CH (**3**) at 60 °C for 16 h afforded triazole **5b** (28.0 mg, 0.106 mmol) as a brown solid. **^1^H NMR** [400 MHz, δ (ppm), CDCl_3_]: 8.26 (s, 1 H, C*H*), 7.79–7.67 (m, 2 H, 2-C*H* + 6-C*H*), 7.31–7.22 (m, 2 H, 3-C*H* + 5-C*H*). **^13^C NMR** [101 MHz, δ (ppm), CDCl_3_]: 163.1 (d, *J* = 250.6 Hz, 4-*C*), 132.7 (1-*C*), 130.8 (*C*–S), 128.6 (*C*H), 128.4 (q, *J* = 309.7 Hz, S*C*F_3_), 123.0 (d, *J* = 8.8 Hz, 2-*C* + 6-*C*), 117.2 (d, *J* = 23.5 Hz, 3-*C* + 5-*C*). **^19^F NMR** [377 MHz, δ (ppm), CDCl_3_]: −42.7 (SC*F*_3_). **FTIR** [$\bar{\nu }$ (cm^−1^)]: 3125, 1520, 1243, 1146, 1122, 839. **HRMS** [ESI (m/z)] calcd for (C_9_H_5_F_4_N_3_S + H)^+^ = 264.02186, found 264.02227 (|Δ| = 1.59 ppm). ***R*_F_**: 0.42 (heptane/AcOEt, 4:1). **Yield**: 85%.

#### 3.4.3. 1-[4-(Trifluoromethyl)phenyl]-4-[(trifluoromethyl)sulfanyl]-1*H*-1,2,3-triazole (**5c**)

According to the general procedure, the reaction of 1-azido-4-(trifluoromethyl)benzene **8c** (250 µL, 0.125 mmol, 0.5 M solution in *^t^*BuOMe) with CF_3_S–C≡CH (**3**) at 50 °C for 16 h afforded triazole **5c** (31.3 mg, 0.10 mmol) as a yellow solid. **^1^H NMR** [400 MHz, δ (ppm), CDCl_3_]: 8.38 (s, 1 H, *CH*), 7.97–7.89 (m, 2 H, 2-C*H* + 6-C*H*), 7.88–7.82 (m, 2 H, 3-C*H* + 5-C*H*). **^13^C NMR** [101 MHz, δ (ppm), CDCl_3_]: 138.8 (1-*C*), 131.9 (q, *J* = 33.4 Hz, 4-*C*), 131.4 (*C*–S), 128.37 (q, *J* = 309.9 Hz, S*C*F_3_), 128.37 (*C*H), 127.5 (q, *J* = 3.7 Hz, 3-*C* + 5-*C*), 123.5 (q, *J* = 272.4 Hz, *C*F_3_), 120.9 (2-*C* + 6-*C*). **^19^F NMR** [377 MHz, δ (ppm), CDCl_3_]: −42.6 (SC*F*_3_), −62.8 (C*F*_3_). **FTIR** [$\bar{\nu }$ (cm^−1^)]: 3117, 1335, 1151, 1106, 844. **HRMS** [ESI (m/z)] calcd for (C_10_H_5_F_6_N_3_S + H)^+^ = 314.01866, found 314.02077 (|Δ| = 2.11 mmu). ***R*_F_**: 0.50 (heptane/AcOEt, 4:1). **Yield**: 80%.

#### 3.4.4. 1-(4-Methylphenyl)-4-[(trifluoromethyl)sulfanyl]-1*H*-1,2,3-triazole (**5d**)

According to the general procedure, the reaction of 1-azido-4-methylbenzene **8d** (250 µL, 0.125 mmol, 0.5 M solution in *^t^*BuOMe) with CF_3_S–C≡CH (**3**) at 70 °C for 48 h afforded triazole **5d** (28.0 mg, 0.108 mmol) as a brown solid. **^1^H NMR** [400 MHz, δ (ppm), CDCl_3_]: 8.26 (s, 1 H, C*H*), 7.64–7.59 (m, 2 H, 2-C*H* + 6-C*H*), 7.38–7.32 (m, 2 H, 3-C*H* + 5-C*H*), 2.44 (s, 3 H, C*H*_3_). **^13^C NMR** [101 MHz, δ (ppm), CDCl_3_]: 140.1 (4-*C*), 134.2 (1-*C*), 130.6 (3-*C* + 5-*C*), 130.3 (*C*–S), 128.46 (q, *J* = 309.7 Hz, S*C*F_3_), 128.45 (*C*H), 120.7 (2-*C* + 6-*C*), 21.3 (*C*H_3_). **^19^F NMR** [377 MHz, δ (ppm), CDCl_3_]: −42.8. **FTIR** [$\bar{\nu }$ (cm^−1^)]: 2967, 1143, 1119, 1039, 817. **HRMS** [ESI (m/z)] calcd for (C_10_H_8_F_3_N_3_S + H)^+^ = 260.04693, found 260.04638 (|Δ| = 2.11 ppm). ***R*_F_**: 0.44 (heptane/AcOEt, 4:1). **Yield**: 86%. Compound **5d** could not be isolated in pure form after column chromatography.

#### 3.4.5. 1-(4-Methoxyphenyl)-4-[(trifluoromethyl)sulfanyl]-1*H*-1,2,3-triazole (**5e**)

According to the general procedure, the reaction of 1-azido-4-methoxybenzene **8e** (250 µL, 0.125 mmol, 0.5 M solution in *^t^*BuOMe) with CF_3_S–C≡CH (**3**) at 50 °C for 72 h afforded a 12:1 mixture of 1,4-/1,5-disubstituted triazole **5e** (7.0 mg, 0.025 mmol) as a yellow–brown solid. **^1^H NMR** [400 MHz, δ (ppm), CDCl_3_]: 8.21 (s, 1 H, C*H*), 7.67–7.62 (m, 2 H, 2-C*H* + 6-C*H*), 7.10–7.01 (m, 2 H, 3-C*H* + 5-C*H*), 3.89 (s, 3 H, OC*H*_3_). **^13^C NMR** [101 MHz, δ (ppm), CDCl_3_]: 160.5 (4-*C*), 130.1 (1-*C*), 129.7 (*C*–S), 128.4 (*C*H), 128.3 (q, *J* = 309.6 Hz, S*C*F_3_), 122.4 (2-*C* + 6-*C*), 115.0 (3-*C* + 5-*C*), 55.7 (O*C*H_3_). **^19^F NMR** [377 MHz, δ (ppm), CDCl_3_]: −42.8. **FTIR** [$\bar{\nu }$ (cm^−1^)]: 2926, 1522, 1261, 1145, 1121, 830. **HRMS** [ESI (m/z)] calcd for (C_10_H_8_F_3_N_3_OS + H)^+^ = 276.04184, found 276.04126 (|Δ| = 2.11 ppm). ***R*_F_**: 0.32 (heptane/AcOEt, 4:1). **Yield**: 20%. The ratio between the corresponding 1,4- and 1,5-regioisomers was not determined from the crude mixtures but from the corresponding purified fractions.

#### 3.4.6. 1-Phenyl-4-[(trifluoromethyl)sulfanyl]-1*H*-1,2,3-triazole (**5f**)

According to the general procedure, the reaction of phenyl azide **8f** (250 µL, 0.125 mmol, 0.5 M solution in *^t^*BuOMe) with CF_3_S–C≡CH (**3**) at 30 °C for 48 h afforded triazole **5f** (22.4 mg, 0.09 mmol) as a brown solid. **^1^H NMR** [400 MHz, δ (ppm), CDCl_3_]: 8.30 (s, 1 H, C*H*), 7.78–7.73 (m, 2 H, 2-C*H* + 6-C*H*), 7.61–7.55 (m, 2 H, 3-C*H* + 5-C*H*), 7.54–7.48 (m, 1 H, 4-C*H*). **^13^C NMR** [101 MHz, δ (ppm), CDCl_3_]: 136.2 (1-*C*), 130.5 (*C–*S, indirect observation), 130.2 (3-*C* + 5-*C*), 129.8 (4-*C*), 128.5 (*C*H), 120.9 (2-*C* + 6-*C*). The carbon signal of SCF_3_ was not observed. **^19^F NMR** [377 MHz, δ (ppm), CDCl_3_]: −42.8. **FTIR** [$\bar{\nu }$ (cm^−1^)]: 2989, 1141, 1120, 1041, 758. **HRMS** [ESI (m/z)] calcd for (C_9_H_6_F_3_N_3_S + H)^+^ = 246.03128, found 246.03124 (|Δ| = 0.15 ppm). ***R*_F_**: 0.36 (heptane/AcOEt, 4:1). **Yield**: 73%.

#### 3.4.7. 1-(3-Chlorophenyl)-4-[(trifluoromethyl)sulfanyl]-1*H*-1,2,3-triazole (**5g**)

According to the general procedure, the reaction of 1-azido-3-chlorobenzene **8g** (200 µL, 0.10 mmol, 0.5 M solution in *^t^*BuOMe) with CF_3_S–C≡CH (**3**) at 30 °C for 16 h afforded triazole **5g** (21.0 mg, 0.075 mmol) as a brown solid. **^1^H NMR** [400 MHz, δ (ppm), CDCl_3_]: 8.32 (s, 1 H, C*H*), 7.81 (td, *J* = 1.9, 0.6 Hz, 1 H, 2-C*H*), 7.67 (dt, *J* = 7.4, 2.0 Hz, 1 H, 6-C*H*), 7.53 (t, *J* = 7.7 Hz, 1 H, 5-C*H*), 7.48 (dt, *J* = 8.1, 1.8 Hz, 1 H, 4-C*H*). **^13^C NMR** [101 MHz, δ (ppm), CDCl_3_]: 137.2 (1-*C* or 3-*C*), 136.1 (1-*C* or 3-*C*), 131.2 (5-*C*), 131.0 (*C*–S), 129.9 (4-*C*), 128.43 (*C*H), 128.39 (q, *J* = 309.8 Hz, S*C*F_3_), 121.1 (2-*C*), 118.8 (6-*C*). **^19^F NMR** [377 MHz, δ (ppm), CDCl_3_]: −42.7. **FTIR** [$\bar{\nu }$ (cm^−1^)]: 3124, 1597, 1155, 1118, 1040, 784. **HRMS** [ESI (m/z)] calcd for (C_9_H_5_F_3_N_3_SCl + H)^+^ = 279.99231, found 279.99238 (|Δ| = 0.26 ppm). ***R*_F_**: 0.40 (heptane/AcOEt, 4:1). **Yield**: 75%.

#### 3.4.8. 1-(3-Methoxyphenyl)-4-[(trifluoromethyl)sulfanyl]-1*H*-1,2,3-triazole (**5h**)

According to the general procedure, the reaction of 1-azido-3-methoxybenzene **8h** (19 mg, 0.13 mmol) with CF_3_S–C≡CH (**3**) at 50 °C for 48 h afforded a 10:1 mixture of 1,4-/1,5-disubstituted triazole **5h** (12.9 mg, 0.047 mmol) as a brown solid. **^1^H NMR** [400 MHz, δ (ppm), CDCl_3_]: 8.28 (s, 1 H, C*H*), 7.45 (t, *J* = 8.2 Hz, 1 H, 5-C*H*), 7.35 (t, *J* = 2.3 Hz, 1 H, 2-C*H*), 7.28 (ddd, *J* = 8.2, 2.1, 0.9 Hz, 1 H, 6-C*H*), 7.03 (ddd, *J* = 8.2, 2.6, 0.9 Hz, 1 H, 4-C*H*), 3.90 (s, 3 H, OC*H*_3_). **^13^C NMR** [101 MHz, δ (ppm), CDCl_3_]: 160.9 (3-*C*), 137.5 (1-*C*), 130.9 (5-*C*), 130.4 (*C*–S), 128.6 (*C*H), 128.5 (q, *J* = 309.6 Hz, S*C*F_3_), 115.6 (4-*C*), 112.6 (6-*C*), 106.7 (2-*C*), 55.9 (O*C*H_3_). **^19^F NMR** [377 MHz, δ (ppm), CDCl_3_]: −42.8. **FTIR** [$\bar{\nu }$ (cm^−1^)]: 2932, 1612, 1146, 1110, 1032. **HRMS** [ESI (m/z)] calcd for (C_10_H_8_F_3_N_3_OS + H)^+^ = 276.04184, found 276.04185 (|Δ| = 0.03 ppm). ***R*_F_**: 0.34 (heptane/AcOEt, 4:1). **Yield**: 36%. The ratio between the corresponding 1,4- and 1,5-regioisomers was not determined from the crude mixtures but from the corresponding purified fractions.

#### 3.4.9. 3-{4-[(Trifluoromethyl)sulfanyl]-1*H*-1,2,3-triazol-1-yl}benzonitrile (**5i**)

According to the general procedure, the reaction of 3-azidobenzonitrile **8i** (20 mg, 0.14 mmol) with CF_3_S–C≡CH (**3**) at 50 °C for 48 h afforded triazole **5i** (15.1 mg, 0.056 mmol) as a brown solid. **^1^H NMR** [400 MHz, δ (ppm), CDCl_3_]: 8.35 (s, 1 H, C*H*), 8.10 (t, *J* = 1.9 Hz, 1 H, 2-C*H*), 8.05 (ddd, *J* = 8.1, 2.3, 1.2 Hz, 1 H, 4-C*H*), 7.81 (dt, *J* = 7.8, 1.3 Hz, 1 H, 6-C*H*), 7.73 (t, *J* = 7.9 Hz, 1 H, 5-C*H*). **^13^C NMR** [101 MHz, δ (ppm), CDCl_3_]: 137.0 (3-*C*), 133.1 (6-*C*), 131.36 (5-*C*), 131.34 (*C*–S), 128.2 (*C*H), 124.8 (4-*C*), 124.0 (2-*C*), 117.1 (*C*N), 114.7 (1-*C*). The carbon signal of SCF_3_ was not observed. **^19^F NMR** [377 MHz, δ (ppm), CDCl_3_]: −42.5. **FTIR** [$\bar{\nu }$ (cm^−1^)]: 2930, 2235, 1109, 1032, 755. ***R*_F_**: 0.17 (heptane/AcOEt, 4:1). **Yield**: 40%.

#### 3.4.10. 1-(2-Methoxyphenyl)-4-[(trifluoromethyl)sulfanyl]-1*H*-1,2,3-triazole (**5j**)

According to the general procedure, the reaction of 1-azido-2-methoxybenzene **8j** (19 mg, 0.13 mmol) with CF_3_S–C≡CH (**3**) at 50 °C for 48 h afforded a 10:1 mixture of 1,4-/1,5-disubstituted triazole **5j** (11.0 mg, 0.04 mmol) as a brown solid. **^1^H NMR** [400 MHz, δ (ppm), CDCl_3_]: 8.48 (s, 1 H, C*H*), 7.86 (dd, *J* = 7.9, 1.7 Hz, 1 H, 6-C*H*), 7.47 (ddd, *J* = 8.3, 7.6, 1.7 Hz, 1 H, 4-C*H*), 7.17–7.10 (m, 2 H, 3-C*H* + 5-C*H*), 3.93 (s, 3 H, OC*H*_3_). **^13^C NMR** [101 MHz, δ (ppm), CDCl_3_]: 150.8 (2-*C*), 132.4 (*C*H), 130.8 (4-*C*), 128.8 (*C*–S), 128.4 (q, *J* = 309.7 Hz, S*C*F_3_), 125.6 (1-*C*), 125.1 (6-*C*), 121.4 (5-*C*), 112.3 (3-*C*), 56.1 (O*C*H_3_). **^19^F NMR** [377 MHz, δ (ppm), CDCl_3_]: −43.1. **FTIR** [$\bar{\nu }$ (cm^−1^)]: 2939, 1604, 1510, 1285, 1256, 1106, 1026, 754. **HRMS** [ESI (m/z)] calcd for (C_10_H_8_F_3_N_3_OS + H)^+^ = 276.04184, found 276.04158 (|Δ| = 0.93 ppm). ***R*_F_**: 0.38 (heptane/AcOEt, 4:1). **Yield**: 31%. The ratio between the corresponding 1,4- and 1,5-regioisomers was not determined from the crude mixtures but from the corresponding purified fractions.

#### 3.4.11. 2-{4-[(Trifluoromethyl)sulfanyl]-1*H*-1,2,3-triazol-1-yl}benzonitrile (**5k**)

According to the general procedure, the reaction of 2-azidobenzonitrile **8k** (18 mg, 0.12 mmol) with CF_3_S–C≡CH (**3**) at 50 °C for 48 h afforded triazole **5k** (9.7 mg, 0.036 mmol) as a brown solid. **^1^H NMR** [400 MHz, δ (ppm), CDCl_3_]: 8.59 (s, 1 H, C*H*), 7.95 (dd, *J* = 8.2, 1.2 Hz, 1 H, 3-C*H*), 7.91 (dd, *J* = 7.8, 1.5 Hz, 1 H, 6-C*H*), 7.87 (td, *J* = 7.8, 1.5 Hz, 1 H, 4-C*H*), 7.68 (td, *J* = 7.7, 1.3 Hz, 1 H, 5-C*H*).**^13^C NMR** [101 MHz, δ (ppm), CDCl_3_]: 137.8 (2-*C*), 134.8 (4-*C*), 134.6 (6-*C*), 131.0 (*C*–S, indirect observation), 130.8 (*C*H), 130.5 (5-*C*), 128.4 (q, *J* = 309.9 Hz, S*C*F_3_), 125.7 (3-*C*), 115.3 (*C*N), 106.9 (1-*C*). **^19^F NMR** [377 MHz, δ (ppm), CDCl_3_]: −42.6. **FTIR** [$\bar{\nu }$ (cm^−1^)]: 2923, 2853, 1520, 1349, 1146, 1103, 1034, 853. **HRMS** [ESI (m/z)] calcd for (C_10_H_5_F_3_N_4_S + H)^+^ = 271.02653, found 271.02598 (|Δ| = 2.00 ppm). ***R*_F_**: 0.14 (heptane/AcOEt, 4:1). **Yield**: 30%.

#### 3.4.12. 1-(2,6-Difluorophenyl)-4-[(trifluoromethyl)sulfanyl]-1*H*-1,2,3-triazole (**5l**)

According to the general procedure, the reaction of 2-azido-1,3-difluorobenzene **8l** (21 mg, 0.14 mmol) with CF_3_S–C≡CH (**3**) at 50 °C for 48 h afforded triazole **5l** (23.9 mg, 0.085 mmol) as a brown solid. **^1^H NMR** [400 MHz, δ (ppm), CDCl_3_]: 8.18 (s, 1 H, C*H*), 7.55 (tt, *J* = 8.6, 6.0 Hz, 1 H, 4-C*H*), 7.22–7.14 (m, 2 H, 3-C*H* + 5-C*H*). **^13^C NMR** [101 MHz, δ (ppm), CDCl_3_]: 156.8 (dd, *J* = 257.5, 2.7 Hz, 2-*C* + 6-*C*), 133.0 (*C*H), 132.3 (t, *J* = 9.7 Hz, 4-C*H*), 130.0 (*C*–S), 128.37 (q, *J* = 309.7 Hz, S*C*F_3_), 113.0–112.7 (m, 3-*C* + 5-*C*). The signal of 1-*C* was not observed. **^19^F NMR** [377 MHz, δ (ppm), CDCl_3_]: −42.82 (SC*F*_3_). **FTIR** [$\bar{\nu }$ (cm^−1^)]: 2919, 1480, 1111, 1032, 1014, 788. **HRMS** [ESI (m/z)] calcd for (C_9_H_4_F_5_N_3_S + H)^+^ = 282.01243, found 282.01226 (|Δ| = 0.63 ppm). ***R*_F_**: 0.21 (heptane/AcOEt, 4:1). **Yield**: 61%.

#### 3.4.13. 4-{4-[(Trifluoromethyl)sulfanyl]-1*H*-1,2,3-triazol-1-yl}pyridine (**5m**)

According to the general procedure, the reaction of 4-azidopyridine **8m** (19 mg, 0.16 mmol) with CF_3_S–C≡CH (**3**) at 70 °C for 16 h afforded triazole **5m** (25.6 mg, 0.104 mmol) as a brown solid. **^1^H NMR** [400 MHz, δ (ppm), CDCl_3_]: 9.10–8.60 (m, 2 H, 2-C*H* + 6-C*H*), 8.44 (s, 1 H, C*H*), 7.79–7.72 (m, 2 H, 3-C*H* + 5-C*H*). **^13^C NMR** [101 MHz, δ (ppm), CDCl_3_]: 150.9 (2-*C* + 6-*C*), 141.4 (4-*C*), 130.7 (*C*–S), 127.2 (q, *J* = 310.0 Hz, S*C*F_3_), 126.7 (*C*H), 112.8 (3-*C* + 5-*C*). **^19^F NMR** [377 MHz, δ (ppm), CDCl_3_]: −42.5. **FTIR** [$\bar{\nu }$ (cm^−1^)]: 3113, 1587, 1510, 1150, 1111, 1037, 845, 706. ***R*_F_**: 0.12 (heptane/AcOEt, 4:1). **Yield**: 65%.

#### 3.4.14. 1-Benzyl-4-[(trifluoromethyl)sulfanyl]-1*H*-1,2,3-triazole (**5n**)

According to the general procedure, the reaction of (azidomethyl)benzene **8n** (22 mg, 0.17 mmol) with CF_3_S–C≡CH (**3**) at 50 °C for 16 h afforded triazole **5n** (27.2 mg, 0.105 mmol) as a brown solid. **^1^H NMR** [400 MHz, δ (ppm), CDCl_3_]: 7.76 (s, 1 H, C*H*), 7.44–7.37 (m, 3 H, 3-C*H* + 4-C*H +* 5-C*H*), 7.31–7.27 (m, 2 H, 2-C*H* + 6-C*H*), 5.58 (s, 2 H, NC*H*_2_). **^13^C NMR** [101 MHz, δ (ppm), CDCl_3_]: 133.7 (1-*C*), 130.3 (*C*–S), 130.2 (*C*H), 129.5 (3-*C* + 5-*C*), 129.4 (4-*C*), 128.42 (q, *J* = 309.5 Hz, S*C*F_3_), 128.37 (2-*C* + 6-*C*), 54.9 (N*C*H_2_). **^19^F NMR** [377 MHz, δ (ppm), CDCl_3_]: −43.0. **FTIR** [$\bar{\nu }$ (cm^−1^)]: 2362, 1984, 1143, 1129, 1044, 716. **HRMS** [ESI (m/z)] calcd for (C_10_H_8_F_3_N_3_S + H)^+^ = 260.04693, found 260.04899 (|Δ| = 2.06 mmu). ***R*_F_**: 0.28 (heptane/AcOEt, 4:1). **Yield**: 62%.

#### 3.4.15. 1-(Adamantan-1-yl)-4-[(trifluoromethyl)sulfanyl]-1*H*-1,2,3-triazole (**5o**)

According to the general procedure, the reaction of 1-azidoadamantane **8o** (22 mg, 0.12 mmol) with CF_3_S–C≡CH (**3**) at 50 °C for 16 h afforded triazole **5o** (26.2 mg, 0.09 mmol) as a pale solid. **^1^H NMR** [400 MHz, δ (ppm), CDCl_3_]: 7.90 (s, 1 H, C*H*), 2.32–2.23 (m, 9 H, adamantyl), 1.88–1.75 (m, 6 H, adamantyl). **^13^C NMR** [101 MHz, δ (ppm), CDCl_3_]: 128.4 (q, *J* = 309.0 Hz, S*C*F_3_), 128.2 (*C*–S), 126.9 (*C*H), 60.9 (1-*C*), 42.8 (2-*C*), 35.8 (4-*C*), 29.4 (3-*C*). **^19^F NMR** [377 MHz, δ (ppm), CDCl_3_]: −43.2. **FTIR** [$\bar{\nu }$ (cm^−1^)]: 2915, 1140, 1109, 1019. **HRMS** [ESI (m/z)] calcd for (C_13_H_16_F_3_N_3_S + H)^+^ = 304.10953, found 304.11024 (|Δ| = 0.71 mmu). ***R*_F_**: 0.48 (heptane/AcOEt, 4:1). **Yield**: 70%.

## 4. Conclusions

In summary, we have developed the first method for the synthesis of ethynyl trifluoromethyl sulfide **3** as the major product, a building block for the synthesis of CF_3_S-containing heterocycles. This method involves one-pot three-step synthesis, which was directly applied to synthesize CF_3_S-containing triazoles via copper-catalyzed 1,3-dipolar cycloaddition reactions with several (hetero)aromatic and aliphatic azides. Currently, the biological properties of these compounds are being evaluated in various assays, and the reactivity of the CF_3_S–C≡CH (**3**) is being further explored with different dipoles.

## Data Availability

The data presented in this study are contained within the article.

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
