# Peer review of "Synthesis of Ethynyl Trifluoromethyl Sulfide and Its Application to the Synthesis of CF3S-Containing Triazoles"

_molecules, 2025, doi:10.3390/molecules30112358_

Round 1

Reviewer 1 Report

Comments and Suggestions for Authors

This manuscript by Rutjes and coworkers reported a method for the synthesis of ethynyl trifluoromethyl sulfide, which serves as a building block for the preparation of CF3S-containing heterocycles. In addition, application of ethynyl trifluoromethyl sulfide in the synthesis of CF3S-containing triazoles by means of a copper-catalyzed 1,3-dipolar cycloaddition reactions with (hetero)aromatic and aliphatic azides have been established. The manuscript is recommended for publication in Molecules after the following minor revisions are made by the authors.

  1. The resolution of Figure and Scheme is not high.
  2. Page 2, a space should be put in “herewith”.
  3. Table 1 and 2, “entry” should be deleted in “entry 1”, “entry 2”, and others.
  4. It is suggested that the footnote 1, 2, 3, … in Tables could be changed to a, b, c, …
  5. Table 2, footnote 1 writes “by using a mixture of KOtBu (1.5 equiv) and NaHMDS (1.7 equiv).”; please double-check the amount of KOtBu is 1.5 equiv or 1.3 equiv.?
  6. Scheme 3 should be cited in manuscript text when the result in Scheme 3 is first discussed.
  7. The use of footnotes 1-6 is not recommended in Scheme 3. The reaction temperature and reaction time could be directly put after product yield in a bracket. For example: 5a, 82% (30 oC, 16 h).
  8. Adjective like (trifluoromethyl)sulfonyl and (trifluoromethyl)thio should not be used as keyword.
  9. Please double-check the calculated yields for products 5j and 5o.
  10. Product 5l, “400” in “19F NMR [400 MHz” should be corrected.

Author Response

Comment 1: The resolution of Figure and Scheme is not high.

Response 1: Thank you for pointing this out. We agree with this comment. This problem was due to a Mac/PC incompatibility. The schemes and figures were redrawn with a different computer and saved with a higher resolution (TIFF files).

Comment 2: Page 2, a space should be put in “herewith”.

Response 2: We are not sure what is meant by this comment. The word “herewith” is only once in the manuscript, it presents spaces at both sides of the word and it is a single word, not “here” and “with”. We have decided to leave it the way it was written.

Comment 3: Table 1 and 2, “entry” should be deleted in “entry 1”, “entry 2”, and others.

Response 3: Thank you for pointing this out. We agree with this comment. Therefore, we have deleted “entry” from all the cells of tables 1 and 2.

Comment 4: It is suggested that the footnote 1, 2, 3, … in Tables could be changed to a, b, c,…

Response 4: We do agree with this comment because we also like it that way, but we have decided to stick to the template given by the journal.

Comment 5: Table 2, footnote 1 writes “by using a mixture of KOtBu (1.5 equiv) and NaHMDS (1.7 equiv).”; please double-check the amount of KOtBu is 1.5 equiv or 1.3 equiv.?

Response 5: Well spotted. We agree with this comment. Therefore, we have made the corresponding change to “1.3 equiv” (the right amount used).

(page 4 and line 151)

Comment 6: Scheme 3 should be cited in manuscript text when the result in Scheme 3 is first discussed.

Response 6: “Scheme 3” is cited when it is first commented in the text. “A total of 12 aromatic azides (8al) with electron-donating or electron-withdrawing groups at the 2-, 3- and 4-positions were used to study the scope of the copper-catalyzed cycloaddition reaction with CF3S–C≡CH (3; Scheme 3).” can be read in the text. We don’t see any other proper location for this call-out.

(page 5, paragraph 2, and line 192)

Comment 7: The use of footnotes 1-6 is not recommended in Scheme 3. The reaction temperature and reaction time could be directly put after product yield in a bracket. For example: 5a, 82% (30 °C, 16 h).

Response 7: Thank you for pointing this out. We agree with this comment. Therefore, we have modified the scheme and included the information the way suggested for all the entries.

(Scheme 3, page 5, line 199)

Comment 8: Adjective like (trifluoromethyl)sulfonyl and (trifluoromethyl)thio should not be used as keyword.

Response 8: We do not understand the reason for this. One of the main points of this research is the incorporation of those groups into organic molecules and those are the names used by the research community. Therefore, we still think it is important to use those two names as keywords to find this article when browsing. Maybe there was some confusion because (trifluoromethyl)sulfonyl was mentioned instead of (trifluoromethyl)sulfanyl.

Comment 9: Please double-check the calculated yields for products 5j and 5o.

Response 9: Thank you for pointing this out. We have recalculated the yields and corrected them in Scheme 3 and in the experimental section. The actual yield of 5j was 31% instead of 30% and the yield of 5o was 70% instead of 72%.

(Scheme 3, page 5, line 199; page 6, paragraph 2, line 218; page 10, paragraph 4, line 440; page 11, paragraph 5, line 491)

Comment 10: Product 5l, “400” in “19F NMR [400 MHz” should be corrected.

Response 10: Thank you for pointing this out. We agree with this comment. Therefore, we have changed “400” to “377”.

(page 11, paragraph 2, and line 462)

Reviewer 2 Report

Comments and Suggestions for Authors

The authors report the synthesis of a new reagent, ethynyl trifluoromethyl sulfide and its use in the click reaction to form triazols containing SCF3 group. This work is a continuation of previous reports from the group focusing on synthesis of SCF3 containing heterocycles, using vinyl SCF3. 

The reported reactions and reagents are novel and of an interest to a synthetic chemist. The manuscript is very written with a high attention to details. The quality of the schemes, however, needs to be improved - schemes are barely legible - I recommend to authors to save them in a different format. The way of presenting experimental data is not always immediately obvious. For example, it was not immediately clear to me how many equivalents was used in Table 2. Maybe it will be better to add - "1.0 equiv. added after x time" instead of just a sign "+"? Scheme 2 will benefit from including yields. It is also not immediately clear to me why compound 3 has not been fully characterized (only H and F NMR shifts are provided). For every new compound, a full characterization is usually necessary, including C NMR and HRMS. Can authors fill this gap and include these data? In general, this is a nice work.

Author Response

Comment 1: The quality of the schemes needs to be improved.

Response 1: Thank you for pointing this out. We agree with this comment. This problem was due to a Mac/PC incompatibility. The schemes and figures were redrawn with a different computer and saved with a higher resolution (TIFF files).

Comment 2: The way of presenting experimental data is not always immediately obvious. For example, it was not immediately clear to me how many equivalents was used in Table 2. Maybe it will be better to add - "1.0 equiv. added after x time" instead of just a sign "+"?

Response 2: We completely agree with this comment (we understand that Table 1 is meant). We have modified the table: we have deleted the plus signs and added “then” in all the entries to make the second addition of base clearer.

(page 3, line 79)

Comment 3: Scheme 2 will benefit from including yields.

Response 3: We’d love to be able to report the yields for Scheme 2, but unfortunately neither compound 2 nor 6 were isolated because of the small scale we worked on. We tried unsuccessfully to isolate compound 6 (as stated in the text), but it decomposed on the attempt.

Comment 4: It is also not immediately clear to me why compound 3 has not been fully characterized (only H and F NMR shifts are provided). For every new compound, a full characterization is usually necessary, including C NMR and HRMS. Can authors fill this gap and include these data? In general, this is a nice work.

Response 4: Compound 3 was not isolated in pure form; we made a stock solution for the subsequent reactions. The carbon atoms could not be seen when the NMR spectra were conducted with the stock solution.

Reviewer 3 Report

Comments and Suggestions for Authors

General Comments:

In this manuscript, Rutjes, F. P. J. T. and coworkers described synthesis of Ethynyl Trifluoromethyl Sulfide from 2-chloroethyl trifluoromethyl sulfide and a combination of KOt-Bu and NaHMDS. The authors studied its application for synthesis of CF3S-containing triazoles using click reaction.
            The manuscript is well written and easy to follow for the most part. The conclusions made by the authors are in accordance with the data presented in the manuscript.

My recommendation is — accept with minor revisions.

Technical Comments:

  1. The authors could report synthesis of 3 as a stepwise but facile, one-pot synthesis without needing any purification. As reported, 3 can be easily synthesized from commercially available starting materials within 6 hours. To me, it is a strength of the manuscript and could be clearly demonstrated in a scheme.
  2. The authors are appreciated for mentioning that the compound 6 was not isolated and generated in-situ. Can authors monitor yields of 2, 6 and 3 by 19F NMR by using internal standard? If yes, it should be included in Table 1.
  3. In Table 2 reaction scheme, the authors should clearly mention reaction conditions on the arrow.
  4. Table 2 is confusing and contain unnecessary information. The authors are recommended to modify the table to show important data, focus on results with Cu-catalyzed reaction conditions with one azide. The impact of pressure and temperature can be added in one or two entries.
  5. In case where methoxy-substituted aryl azides are used, the reaction seems to give 1,4-regio-isomer in higher yield than 1,5. There is clearly electronic influence on the regioselectivity. The authors could mention these products in Scheme 3, as these products are still useful.

Author Response

Comment 1: The authors could report synthesis of 3 as a stepwise but facile, one-pot synthesis without needing any purification. As reported, 3 can be easily synthesized from commercially available starting materials within 6 hours. To me, it is a strength of the manuscript and could be clearly demonstrated in a scheme.

Response 1: This is a great idea. We have added “three steps (6 h of work!)” on top of the arrow to emphasize this feature of our work on the synthesis of compound 3 from commercially starting materials (compound 1; already included in Scheme 1).

(page 2, line 48)

Comment 2: The authors are appreciated for mentioning that the compound 6 was not isolated and generated in-situ. Can authors monitor yields of 26 and 3 by 19F NMR by using internal standard? If yes, it should be included in Table 1.

Response 2: This is a great idea. Unfortunately, we did not use any internal standard.

Comment 3: In Table 2 reaction scheme, the authors should clearly mention reaction conditions on the arrow.

Response 3: Agree. We have, accordingly, changed the scheme and included the reaction conditions.

(page 4, line 150)

Comment 4: Table 2 is confusing and contain unnecessary information. The authors are recommended to modify the table to show important data, focus on results with Cu-catalyzed reaction conditions with one azide. The impact of pressure and temperature can be added in one or two entries.

Response 4: We appreciate the reviewer’s feedback regarding Table 2. However, we respectfully consider that all the results presented are important for providing a

comprehensive understanding of the reaction scope and limitations. In particular, the inclusion of data beyond the Cu-catalyzed conditions with different azides allows for a clearer comparison and highlights the influence of various parameters such as pressure, temperature, and time. Therefore, we would prefer to retain the complete table as it stands.

Comment 5: In case where methoxy-substituted aryl azides are used, the reaction seems to give 1,4-regio-isomer in higher yield than 1,5. There is clearly electronic influence on the regioselectivity. The authors could mention these products in Scheme 3, as these products are still useful.

Response 5: We agree with this comment. Therefore, we have included the ratios between the 1,4- and 1,5-disubstituted triazoles in Scheme 3 to make clear the formation of the 1,5-isomer. Now it can be read “1,4/1,5 12:1” in the cell for compound 5e, “1,4/1,5 10:1” in the cell for compound 5h, and “1,4/1,5 10:1” in the cell for compound 5j.

(page 5, line 199)

Reviewer 4 Report

Comments and Suggestions for Authors

The manuscript reports the preparation of ethynyl trifluoromethyl sulfide and employed it in CuAAC reaction for the synthesis of CF3S-containing heterocycles. However, in my opinion, this work does not present any conceptual novelty to deserve publication in the journal Molecules, since the synthetic concept (synthesis of SCF3-alkyene and click reaction) has been broadly reported by several authors. I recommend that the author consider some issues highlighted below for a future scientific publication:

  1. The author should provide characterization data and spectra for the synthesized compounds. The reviewer did not find the supplementary materials in the submission system.
  2. Considering the importance of selenium-containing compounds, the reviewer suggests that the authors include an example of a selenium-containing compound (CF3SeC≡CH).
  3. Some images and Schemes are unclear. Please submit high-resolution versions.
  4. Authors should seriously revise the introduction and include references related to more recent work on the synthesis of CSF3-containing triazoles that were not cited in the manuscript such as: Angew. Chem. Int. Ed. 2013, 52, 10814 –10817; Advanced Synthesis & Catalysis 2019, 361(3), 469-475
  5. In Table 1, entries 1-2, the notation for base equivalents (e.g., "2.0 + 1.0") is unclear.
  6. In conclusion: The sentence “which was directly applied to synthesize the first class of CF3S-containing triazoles via copper-catalyzed 1,3-dipolar cycloaddition reactions……” should be revised. Unless the authors are absolutely certain, it is preferable to avoid “first” language. Actually, the synthesis of CF3S-containing triazoles has been reported by Billard et al in 2013 (Angew. Chem. Int. Ed. 2013, 52, 10814 –10817)

Author Response

Comment 1: The author should provide characterization data and spectra for the synthesized compounds. The reviewer did not find the supplementary materials in the submission system.

Response 1: We are terribly sorry to read this. The above-mentioned material was uploaded alongside the rest of the documents. Something may have gone wrong. We will reupload all this material.

Comment 2: Considering the importance of selenium-containing compounds, the reviewer suggests that the authors include an example of a selenium-containing compound (CF3SeC≡CH).

Response 2: We consider this suggestion a good idea, but unfortunately the corresponding selenium compound to our core starting material (CF3SeC≡CH) has not been synthesized yet (Reaxys’ search). We think that the inclusion of other selenium compounds is out of the scope of our work.

Comment 3: Some images and Schemes are unclear. Please submit high-resolution versions.

Response 3: Thank you for pointing this out. We agree with this comment. This problem was due to a Mac/PC incompatibility. The schemes and figures were redrawn with a different computer and saved with a higher resolution (TIFF files).

Comment 4: Authors should seriously revise the introduction and include references related to more recent work on the synthesis of CSF3-containing triazoles that were not cited in the manuscript such as: Angew. Chem. Int. Ed. 2013, 52, 10814 –10817; Advanced Synthesis & Catalysis 2019, 361(3), 469-475.

Response 4: We consider making our literature search more comprehensive a good idea. We appreciate it. The first reference was already included in our publication (ref. 10), and we have included the second suggested reference as reference 38. We placed it where the 1,4-/1,5-regioselectivity was discussed. Now it can be read: “For 1,5-regioselective cycloadditions of azides and CF3S–C≡C–R, see: Song, W.; Zheng, N.; Li, M.; He, J.; Li, J.; Dong, K.; Ullah, K.; Zheng, Y. Rhodium(I)-catalyzed regioselective azi-internal alkynyl trifluoromethyl sulfide cycloaddition and azide-internal thioalkyne cycloaddition under mild conditions. Adv. Synth. Catal. 2019, 361, 469–475.”.

(page number 14 and lines 621–623)

Comment 5: In Table 1, entries 1-2, the notation for base equivalents (e.g., "2.0 + 1.0") is unclear.

Response 5: We completely agree with this comment. We have modified the table: we have deleted the plus signs and added “then” in all the entries to make the second addition of base clearer.

(page 3, line 79)

Comment 6: In conclusion: The sentence “which was directly applied to synthesize the first class of CF3S-containing triazoles via copper-catalyzed 1,3-dipolar cycloaddition reactions……” should be revised. Unless the authors are absolutely certain, it is preferable to avoid “first” language. Actually, the synthesis of CF3S-containing triazoles has been reported by Billard et al in 2013 (Angew. Chem. Int. Ed. 2013, 52, 10814 –10817)

Response 6: Thank you for pointing that out. We agree with this comment. Therefore, we have deleted “the first class of”. Now it can be read “which was directly applied to synthesize CF3S-containing triazoles via copper-catalyzed 1,3-dipolar cycloaddition reactions”.

(page 12, line 499)

Round 2

Reviewer 4 Report

Comments and Suggestions for Authors

In my previous analysis, I commented the novelty of this work. While the authors have made several improvements, my assessment remains unchanged. Therefore, I am not in favor of publishing this manuscript in Molecules.